# OpenReview forum: "Welfare Diplomacy: Benchmarking Language Model Cooperation"
_ICLR.cc/2024/Conference — Submitted to ICLR 2024_

### Official Review · Reviewer_A4aM · 2023-10-26

**Soundness:** 3 good
**Presentation:** 3 good
**Contribution:** 3 good
**Rating:** 6
**Confidence:** 3

**Summary:**

The authors aim to promote societal safety by assisting researchers in developing and assessing multi-agent AI systems. They propose a new benchmark called Welfare Diplomacy for measuring the cooperative capabilities of AI systems., and introduce a general-sum variant of the zero-sum board game Diplomacy, where players must balance military conquest and domestic welfare. They implement the rules of Welfare Diplomacy using an open-source Diplomacy engine and construct baseline agents using zero-shot prompted language models.

**Strengths:**

1.	The authors introduce Welfare Diplomacy (WD) and provide an implementation in an open-source Diplomacy library.
2.	This paper provides theoretical and empirical evidence highlighting the benefits of WD compared to the existing benchmark, Zero-Sum Diplomacy (SD).
3.	The authors develop a language model (LM) scaffolding system to create competent zero-shot baseline agents for WD.

**Weaknesses:**

1. Pareto-efficient equilibria are often not stable，and there may be various factors that can lead to deviations from the equilibrium, such as imperfect information, externalities, or strategic behavior. These deviations can disrupt the equilibrium and lead to a new outcome that is not Pareto-efficient.
2. It is challenging to attain Pareto-efficient equilibria, and how to achieve optimal Nash welfare remains unclear.

**Questions:**

1.	As this paper aims to enhance societal safety by aiding researchers in the development and evaluation of multi-agent AI systems, could you please provide examples that illustrate the potential benefits of using benchmarks in real-world scenarios?
2.	Despite the existence of multiple Pareto-efficient Nash Equilibria (NEs), they often display instability, particularly in complex or realistic scenarios. How can we effectively tackle this challenge?

---

> ### Author Response · Authors · 2023-11-15
> **Response to Reviewer A4aM**
>
> Dear Reviewer A4aM, we appreciate your constructive feedback and your recognition of our contribution to the field with the introduction of Welfare Diplomacy. Your acknowledgment of the theoretical grounding and empirical evidence supporting our work is very encouraging to us. We have made the following revisions and comments in response to your review:
>
> 1.  “Pareto-efficient equilibria are often not stable, and there may be various factors that can lead to deviations from the equilibrium, such as imperfect information, externalities, or strategic behavior. These deviations can disrupt the equilibrium and lead to a new outcome that is not Pareto-efficient.”, “It is challenging to attain Pareto-efficient equilibria, and how to achieve optimal Nash welfare remains unclear.”, “Despite the existence of multiple Pareto-efficient Nash Equilibria (NEs), they often display instability, particularly in complex or realistic scenarios. How can we effectively tackle this challenge?” → We fully agree with the remarks about the instability of Pareto optimal NE, and the difficulty of attaining such profiles. There are a few things to note about this:
>     1.  One hope is that having benchmarks like ours will facilitate research that makes it more likely that we can construct agents that do play Pareto optimal Nash equilibria.
>     2.  We believe there is a need for both more robust solution concepts (e.g., performance against diverse distributions of agents) and the development of environments that are more realistic among the dimensions you mention (e.g., exhibiting noise and private information). We’ve added some remarks about this to our Conclusion, in red.
>     3.  Pareto-optimal Nash equilibrium is a standard operationalization of rational cooperation, and so this is a limitation shared by much of the literature on game theory and multi-agent systems.
> 2.  “As this paper aims to enhance societal safety by aiding researchers in the development and evaluation of multi-agent AI systems, could you please provide examples that illustrate the potential benefits of using benchmarks in real-world scenarios?” → It is true that there are often gaps between benchmarks and real-world applications (e.g., https://aclanthology.org/2021.acl-long.81/). At the same time, it is difficult to improve systems if we do not know what to measure. We aim to push forward the work of measuring cooperative capabilities; we do not think that our Welfare Diplomacy benchmark in itself will inform a particular real-world application. That said, here are some examples of benchmarks that inform real-world scenarios: Claude 2’s system card includes evaluations on benchmarks (e.g., BBQA) which informed deployment (https://www-files.anthropic.com/production/images/Model-Card-Claude-2.pdf); Anthropic’s “responsible scaling policy” requires evaluations for safety (https://www.anthropic.com/index/anthropics-responsible-scaling-policy); OpenAI spent six months after the training of GPT-4 to assess safety concerns, partially informed by benchmarks (https://cdn.openai.com/papers/gpt-4-system-card.pdf).
>
> Many thanks,
>
> The Authors

---

### Official Review · Reviewer_zuKK · 2023-10-27

**Soundness:** 3 good
**Presentation:** 2 fair
**Contribution:** 2 fair
**Rating:** 5
**Confidence:** 4

**Summary:**

This paper introduces Welfare Diplomacy, a variant of the game of Diplomacy that incorporates the balancing of military conquest and domestic welfare. The authors evaluate the proposed variant by developing language model-based agents and comparing different state-of-the-art language models.

**Strengths:**

1. The game of diplomacy is an important challenge in multi-agent research, and the concept of welfare diplomacy is interesting.
2.  The paper effectively explains the differences between the proposed game and existing benchmarks. By making two modifications to the game rules, the nature of the game has been altered, incentivizing players to pursue peace and promoting cooperation.
3. The proposed game and prompts are open-sourced, and experimental results are extensive.

**Weaknesses:**

1. Some arguments regarding the motivations of welfare diplomacy lack rigor and may be questionable. It has been repeatedly claimed in the paper that "While Standard Diplomacy (SD) has features that make it interesting as an environment for cooperative AI research, it is zero-sum and incentivizes the development of cooperation-undermining capabilities" and `In contrast to SD, WD is general-sum'.  However, it has been pointed out in [1] that "In Diplomacy, seven players... coordinate their actions to both cooperate and compete with each other," suggesting that standard diplomacy is not necessarily a zero-sum game. If standard diplomacy were indeed zero-sum, cooperation would not be involved, similar to chess and heads-up poker.

2. The theoretical results are hard to interpret. It would be helpful to clarify the meaning of $\pi^k$ as a NE and how Theorem 1 relates to the main claim.

3. Some important technical details lack clarity. The terms "zero-shot prompted language model", "zero-shot baseline" and "zero-shot evaluations" are used throughout the paper without any specific explanation. Additionally, it would be helpful to provide justification for constructing the exploiter in the paper. Does there exist any agent that has better exploitative power?

[1] "Human-level play in the game of Diplomacy by combining language models with strategic reasoning", Science 2022.

**Questions:**

I would like to see responses to the aforementioned weaknesses.

In addition, I have a question about the metric "basic proficiency". Currently, it is the mean of "the rate of model outputs that are valid JSON and thus able to be parsed without error, the rate of submitted orders that are valid possible orders, and the fraction of global SCs owned by any player and not left neutral." While the first two are understandable, I don't understand why the fraction of global SCs should be considered as an aspect of 'basic proficiency'. To me, it is more like a metric about social welfare.

---

> ### Author Response · Authors · 2023-11-15
> **Response to Reviewer zuKK**
>
> Dear Reviewer zuKK, we are thankful for your comments on the significance of Diplomacy in multi-agent research and your interest in our Welfare Diplomacy variant. Your acknowledgment of our clear explanation of the game differences and the value of open-sourcing our code and results is greatly appreciated! We thank you for your insightful comments and have made several clarifications or revisions to address the points you raised:
> 1. “suggesting that standard diplomacy is not necessarily a zero-sum game. If standard diplomacy were indeed zero-sum, cooperation would not be involved, similar to chess and heads-up poker.” → Yes, it is true that Standard Diplomacy also involves cooperation (between subsets of players). When we say that it is a "zero-sum game", we mean this in the technical sense: The sum of players' scores is the same for any policy profile (because a player's utility is the indicator of whether they won). Although SD can involve cooperation, the fact that there is a single winner means that it has a few limitations as a benchmark for cooperation. In particular, it does not exhibit opportunities for (global) rational cooperation (where “global” means between all players) and skilled play is not differentially (globally) cooperative, as we discuss in Section 2.1.
> 2. “The theoretical results are hard to interpret. It would be helpful to clarify the meaning of  as a NE and how Theorem 1 relates to the main claim.” → We agree that the relevance of Theorem 1 to our main claims should be clarified. We have now edited the paragraph directly after Theorem 1 (see red text), explicitly pointing out the connection to our criterion (A), that there should exist opportunities for global, rational cooperation. In particular, Theorem 1 shows that, in the toy environment, there are Nash equilibria that Pareto-dominate others, and that these Pareto-dominating equilibria require more cooperative capabilities. We also already discussed the connection to Theorem 1 in Section 2.2; we have now added a forward reference to Theorem 1 to make this clear.
> 3. “The terms "zero-shot prompted language model", "zero-shot baseline" and "zero-shot evaluations" are used throughout the paper without any specific explanation.” → By zero-shot, we mean that no examples of gameplay are provided to the model. We have added this clarification to the beginning of Section 4 in red.
> 4. “Additionally, it would be helpful to provide justification for constructing the exploiter in the paper. Does there exist any agent that has better exploitative power?” → In addition to the details provided in the section “exploitability reveals our agents don’t deter deviators”, we have additional details in Appendix B.3. In short, we designed our exploiters based on our intuitions in playing WD. There are likely to be better exploiters as we did not design for optimal exploitation. Our goals with the exploiter experiments were to (i) present a proof of concept for the measurement of exploitability, which is a key property to measure alongside social welfare, and (ii) get a sense for the exploitability of our models, for which these exploiters were sufficient (as they clearly demonstrate that our agents are highly exploitable).
> 5. “I don't understand why the fraction of global SCs should be considered as an aspect of 'basic proficiency'. To me, it is more like a metric about social welfare.” → This metric is about using all the available resources on the board. It’s true that some partitions of the board might involve agreements to leave certain SCs neutral, but in practice, we observed that less capable or worse-prompted models would just not try much to capture free SCs. We wanted a component of the metric to check for this possibility.
>
> Many thanks,
>
> The Authors

---

> > ### Comment · Reviewer_zuKK · 2023-11-22
> >
> > I suggest the author revise their arguments about standard Diplomacy in the paper. The current ones are misleading, as detailed in my review comment. While classifying standard Diplomacy as a zero-sum game may create a stark contrast with existing studies, this oversimplification does not provide an accurate basis for assessing the paper's originality and significance.
> >
> > Regarding other concerns, the response and the revision have addressed them appropriately.

---

> > > ### Author Response · Authors · 2023-11-22
> > > **Response 2 to Reviewer zuKK**
> > >
> > > We thank the reviewer for their engagement with our revisions!
> > >
> > > We acknowledge that the use of the phrase "zero-sum" may misleadingly suggest that standard Diplomacy does not involve any cooperation. We have now revised the draft so that it does not refer to standard Diplomacy as “zero-sum”. Wording replacing the "zero-sum" terminology can be seen in red in the abstract, introduction, and Appendix A. Note that Sections 2.1 and 2.2 contain a number of arguments for the advantages of Welfare Diplomacy over standard Diplomacy as an environment for cooperative AI research, without reference to standard Diplomacy being “zero-sum”.

---

### Official Review · Reviewer_R7WR · 2023-10-31

**Soundness:** 3 good
**Presentation:** 3 good
**Contribution:** 3 good
**Rating:** 8
**Confidence:** 4

**Summary:**

This paper proposes Welfare Diplomacy (WD), a variant of Diplomacy that considers more about agent cooperation. The paper offers
(1) Motivation and illustration of environment design
(2) Nash equilibrium analysis of WD
(3) Experiments that benchmark different LLM models' Nash Welfare and exploitability.

**Strengths:**

Overall I think this is a great paper, the strengths can be addressed as follows:
(1) The paper is clearly written and easy to follow.
(2) The paper proposes a new environment variant to benchmark the agent cooperation ability and clearly illustrate the motivation.
(3) The paper offers a theoretical analysis of its proposed environment and verifies the reasonability of the proposed environment.
(4) The experiments successfully help benchmark the agent cooperation ability.

**Weaknesses:**

The weaknesses are summarized as follows:

(1) The author can try to include more experiment results and ablation studies such as prompt sensitivity, hyperparameter effects, etc.
(2) The author should try to incorporate human-LLM mixed experiments to see how human engagement can influence LLM performance.
(3) Some human analysis of LLM's policy should be conducted to better understand LLM's performance.

**Questions:**

I have the following questions and suggestions:
1. There exist several typos in the paper, e.g. in the first line of section 3 certain NEs certain NEs.
2. Can the authors elaborate more on how the theoretical analysis simplifies the real WD game in section 3.2.1 and theorem 1 and what is the gap between the theoretical analysis and the real WD game?
3. I recommend the author slightly modify the title as the cooperation discussed in the paper is the cooperation in a general-sum game. It can help distinguish itself from the fully cooperative setting.
4. From my perspective, what differentiates the current LLM agent from the previous agent is the ability of the agent to communicate with other agents using language. As shown in the paper, there exist some Pareto efficient policies theoretically. I am a little bit worried about, why bother LLM to do such thing if we can theoretically derive the optimal action (I understand this is a game of language so a language encoder is necessary, but you can also train a language-based RL agent to purely output action). What do the authors think the language communication here can help? My first thought is that communication here can be used during the bargain game and help the equilibrium selection. Can the language help in some other cases (like helping policies but it again fails in the case if you can theoretically derive some optimal action)?
5. It seems most models, except advanced LLMs like GPT-4, cannot have policies that are significantly better than the random baseline, what could be the reasons?

---

> ### Author Response · Authors · 2023-11-15
> **Response to Reviewer R7WR**
>
> Dear Reviewer R7WR, thank you for your evaluation of our paper. Your appreciation for the clarity of our writing and the introduction of a new benchmark for agent cooperation has been encouraging! We also value your recognition of our efforts in providing a theoretical analysis and experimental validation for our environment. We have incorporated several changes in response to your comments:
> 1. “more experiment results and ablation studies such as prompt sensitivity, hyperparameter effects, etc.” → See Appendices B, C. We have also added a small prompt ablation experiment on GPT-4 in Figure 8 of section C.5 in addition to the existing larger prompt ablation on Claude 1.2 in Figure 7.
> 2. “human-LLM mixed experiments to see how human engagement can influence LLM performance” → The suggestion is interesting! We are excited about pursuing this direction in future work; for this work, we were financially constrained and had to narrow our focus.
> 3. “Some human analysis of LLM's policy” → We have already included such analysis in Appendix D, E. We have now also added additional analysis of WDAgent(M) for each of the models M we looked at in Appendix C.4.
> 4. “There exist several typos in the paper, e.g. in the first line of section 3 certain NEs certain NEs.” → Thank you for pointing that out! We went through it once more and corrected typos.
> 5. “Can the authors elaborate more on how the theoretical analysis simplifies the real WD game in section 3.2.1 and theorem 1 and what is the gap between the theoretical analysis and the real WD game?” → We added a description of how the toy version of WD differs from the real version (see new text at the beginning of Section G in red). In short, our toy version has fewer states and more symmetries between players - which makes it much more tractable to analyze - and is centered around a bargaining problem that captures the core challenge of WD.
> 6. “I recommend the author slightly modify the title” → To our knowledge, 'cooperation' is commonly understood as applying to both purely cooperative and mixed-motive games. For example, this is how it is meant in 'cooperative AI' (https://arxiv.org/abs/2012.08630), a subfield we consider our paper to be a part of. In combination with the fact that making the title more precise might require making it too long, we currently think it's best to stick with the present title.
> 7. “I am a little bit worried about, why bother LLM to do such thing if we can theoretically derive the optimal action (I understand this is a game of language so a language encoder is necessary, but you can also train a language-based RL agent to purely output action). What do the authors think the language communication here can help?” → Our interpretation of this question is, why bother to measure the behavior of LLMs If we can build systems to cooperate to begin with? Assuming this interpretation, we think that an increasingly likely world is one with different LM-based systems integrated into society and not built to coordinate with each other by default, as we discuss in the first paragraph of the introduction. We think it is valuable both to prepare for the world we describe and also to work to build another world, such as by building systems that can cooperate optimally. Please correct us if we have misinterpreted the question.
> 8. “My first thought is that communication here can be used during the bargain game and help the equilibrium selection” → Yes, precisely! Even setting aside whether you need language for strong performance, having this environment allows us to measure the properties of LMs that we're interested in.
> 9. “It seems most models, except advanced LLMs like GPT-4, cannot have policies that are significantly better than the random baseline, what could be the reasons?” → Thank you for prompting us to look at this more closely. As part of the newly added Appendix C.4 (in red), we have provided commentary on some of the qualitative features of LLM policies that determine their performance. (Briefly: Claude-1.2 gets very low root mean Nash welfare because in several games, some players hold their units throughout the game and therefore get 0 Welfare Points. GPT-3.5 and Llama-2 each tend not to acquire many more SCs, but do sometimes disband units and therefore acquire some WPs.)
>
> Many thanks,
>
> The Authors

---

### Public Comment · ~Guohao_Li1 · 2023-11-14
**Suggesting related work**

The authors introduce Welfare Diplomacy which emphasizes balancing military conquest and domestic welfare, creating a general-sum environment where players optimize for Welfare Points (WPs) instead of outright victory. The authors propose WD as a robust benchmark for evaluating and fostering cooperative capabilities in AI systems. Key contributions include the formulation of WD rules, implementation in an open-source Diplomacy engine, construction of zero-shot prompted language model (LM) baseline agents, and experiments showing that state-of-the-art models like GPT-4 achieve high social welfare but are exploitable. WD introduces a novel approach to benchmarking cooperative AI, focusing on balancing military and welfare aspects, which is a significant deviation from traditional zero-sum games. The thorough implementation of WD, including its integration into an existing Diplomacy engine and the creation of baseline agents, demonstrates a high level of technical proficiency. The paper provides empirical evidence showing how WD can be used to measure cooperative capabilities in AI systems, contributing to the field of cooperative AI.

Thanks for the commendable work! It could also be beneficial to discuss prior work on multi-LLM agents for the study of cooperative AI [1].

[1] Li, Guohao, Hasan Abed Al Kader Hammoud, Hani Itani, Dmitrii Khizbullin, and Bernard Ghanem. "CAMEL: Communicative Agents for" Mind" Exploration of Large Language Model Society." NeurIPS 2023

---

> ### Author Response · Authors · 2023-11-15
> **Response to Guohao Li**
>
> Dear Guohao Li,
>
> Thank you for your comment and the reference to your work on multi-LLM agents for the study of cooperative AI! We appreciate your suggestion and will take a look at this work.
>
> Best regards,
>
> The Authors

---

### Author Response · Authors · 2023-11-15
**Author Response Summary**

Dear ICLR community and reviewers,

We would like to express our deepest gratitude for the insightful feedback and constructive criticism provided during the review process. Your comments have been very useful for refining our work! We have carefully considered each point raised and have made several revisions (marked in red) to our manuscript to address the concerns and suggestions.

In our revised submission, we expanded our experimental results, including an additional GPT-4 prompt ablation experiment in Figure 8, and we provided additional qualitative analysis of model behaviors across all models in Section C.4. We have also clarified theoretical aspects by elaborating on the simplifications made in our analysis and their implications for the real Welfare Diplomacy game in Section G. Furthermore, we have provided explicit explanations for some technical terms and concepts that were previously unclear.

Additionally, we discussed in our comments below the stability of Pareto-efficient equilibria and how they might be affected in complex scenarios. We have also clarified the meaning of our metrics, particularly the “Basic Proficiency” metric, and have corrected typographical errors in our manuscript.

We believe these revisions have significantly strengthened our paper and hope that the changes meet the reviewers' expectations. We look forward to any further feedback and are excited about the potential impact of our work on the development and evaluation of multi-agent cooperative AI systems.

Thank you once again for your valuable contributions to improving our research.

The Authors

---

### Meta-Review · Area_Chair_sGXm · 2023-12-11

**Metareview:**

This paper proposes Welfare Diplomacy (WD), a variant of Diplomacy that considers more about agent cooperation. The reviewers have deemed the actions of this study as clear-cut and its experiments robust, providing significant insights into the research of multi-agent systems. The authors conducted trials on the environment using agents based on LLMs and open-sourced the code. Most concerns raised by the reviewers have been addressed seriously, and corresponding modifications were made in the manuscript. However, their arguments concerning standard Diplomacy are seen as flawed in the eyes of the reviewer. AC believes this will undermine the contribution and intentions of the article. Some important comments have not been adequately addressed in the text. For example, the limitation of using benchmarks in real-world scenarios, for which the author explained the reasons in their rebuttal. AC would prefer if this were delineated in the paper, as this would facilitate future researchers in better assessing the study's merit.

Based on the above, AC chose to reject this paper as it does not meet the standards required for acceptance at ICLR.

**Justification For Why Not Higher Score:**

N/A

**Justification For Why Not Lower Score:**

N/A

---

### Decision · Program_Chairs · 2024-01-16

Reject